# Frequency of Detection of Respiratory Pathogens in Clinically Healthy Show Horses Following a Multi-County Outbreak of Equine Herpesvirus-1 Myeloencephalopathy in California

**DOI:** 10.3390/pathogens11101161

**Published:** 2022-10-08

**Authors:** Abigail Wilcox, Samantha Barnum, Cara Wademan, Rachel Corbin, Edlin Escobar, Emir Hodzic, Stephen Schumacher, Nicola Pusterla

**Affiliations:** 1Department of Medicine and Epidemiology, School of Veterinary Medicine, University of California, Davis, CA 95616, USA; 2United States Equestrian Federation, Lexington, KY 40511, USA

**Keywords:** EHM outbreaks, respiratory pathogens, surveillance, silent shedders, equids, qPCR

## Abstract

Actively shedding healthy horses have been indicated as a possible source of respiratory pathogen outbreak, transmission, and spread. Using nasal swabs from clinically healthy sport horses submitted for qPCR testing after an outbreak of equine herpesvirus-1 (EHV-1) myeloencephalopathy (EHM) in the spring of 2022, this study aimed to identify the rate of clinically healthy horses shedding common and less characterized respiratory pathogens within the sport horse population to better understand their role in outbreaks. Swabs were collected during a required quarantine and testing period, according to the United States Equestrian Federation (USEF), and showed return-to-competition requirements. Common respiratory pathogens, such as equine influenza virus (EIV), EHV-4, and equine rhinitis B virus (ERBV), were found at low but stable frequencies within previously reported ranges, whereas EHV-1 and *Streptococcus equi* subspecies *equi* (*S. equi*) were found at or above previously reported frequencies. Less characterized respiratory pathogens, such as EHV-2, EHV-5, and *S. equi* subspecies *zooepidemicus* (*S. zooepidemicus*), were found within previously reported ranges. Common respiratory pathogens, especially EHV-1 following the multiple EHM outbreaks, were found to be circulating in clinically healthy sport horse populations, reflecting their silent transmission. The strategy of quarantine and EHV-1 qPCR testing of clinically healthy horses was successful at eliminating additional EHM outbreaks and facilitating safe return to competition with no reported respiratory disease outbreaks following the subsequent shows in California.

## 1. Introduction

Outbreaks of respiratory infections at horse shows have a negative impact on the equine industry, not only through the cancelation of events but also through the costs associated with treating diseased horses, lost training time, and quarantine/isolation protocols [1]. The reasons behind respiratory outbreaks are often multifactorial and difficult to understand [2]. Subclinical infections with respiratory pathogens have been indicated as a possible explanation for outbreaks of disease [3], and horses shedding virus without clinical disease (silent shedder) have been identified as possible sources of transmission and spread [4]. The rate of silent shedders for respiratory pathogens is highly dependent on population age and use, as well as the specific pathogen tested, with reported frequencies ranging from 0 to 4% [2,3,5,6,7,8,9,10]. The stress of transport and competition may increase susceptibility to respiratory pathogens or even allow for latent viral infections to become active again [7,11]. This is magnified in populations of young sport horses, known to experience a higher susceptibility to respiratory pathogens [12].

Nasal swab testing using qPCR is both an accurate and sensitive way to assay samples for a variety of respiratory pathogens in clinically and subclinically infected horses [13]. Therefore, the aim of the present study was to investigate the detection frequency of respiratory pathogens in healthy show horses through the use of qPCR in an attempt to understand the risk of silent transmission through subclinical shedders following multi-county EHV-1 outbreaks during the spring of 2022 in California. The outbreak was associated with 105 horses with fever and respiratory signs and 11 EHM cases originating from 21 different premises located in 10 counties. The study focused on the testing of nasal secretions for common respiratory pathogens (equine influenza virus (EIV), equine herpesvirus type 1 (EHV-1), EHV-4, equine rhinitis A virus (ERAV), ERBV, and *Streptococcus equi* subspecies *equi* (*S. equi*)) and less characterized pathogens (EHV-2, EHV-5, and *Streptococcus equi* subspecies *zooepidemicus* (*S. zooepidemicus*)) collected from clinically healthy show horses prior to return to competition.

## 2. Results

A total of 639 nasal swabs from 484 clinically healthy sport horses were submitted for routine EHV-1 diagnostic testing to the Real-Time PCR Research and Diagnostics Core Facility at the School of Veterinary Medicine, University of California at Davis. In total, 340 horses had one swab submitted, 133 horses had two swabs submitted, and 11 horses had three swabs submitted. At the time of sample collection, horses were located in California (406 horses), Washington (45 horses), Nevada (19 horses), Arizona (13 horses), and Oregon (1 horse).

Seventy-nine swabs from 47 horses were positive for a common respiratory pathogen (Table 1). The frequency of respiratory pathogen detection was as follows: 24 swabs from 21 horses were qPCR positive for EHV-1; 18 swabs from 16 horses were qPCR positive for *S. equi*; 9 swabs from 9 horses were qPCR positive for EHV-4; 8 swabs from 6 horses were qPCR positive for ERBV; and 4 swabs from 4 horses were qPCR positive for EIV. All nasal swabs tested qPCR negative for ERAV. Less characterized respiratory pathogens were also present. In total, 215 swabs from 183 horses tested qPCR positive for EHV-5; 209 swabs from 180 horses tested qPCR positive for EHV-2; and 188 swabs from 176 horses tested qPCR positive for *S. zooepidemicus*. The frequency of common respiratory pathogens peaked during week 3, with 28 of 156 (17.9%) swabs testing qPCR positive. Thereafter, the frequency of detection for common respiratory pathogens declined rapidly and reached 8.8% during week 5. The frequency of detection for less characterized respiratory pathogens remained similar among the various study weeks. 

Clusters of EHV-1, EHV-4, *S. equi*, and EIV occurred with at least two qPCR-positive swabs in the same barn during the same week (Table 2). Clusters occurred in barns F, H, M, and AA for EHV-1; barns J, M, and AC for EHV-4; barns K and Q for *S. equi*; and barns G and H for EIV. ERBV did not show more than one qPCR-positive sample in a barn. 

The detection rate for common respiratory pathogens for horses with multiple swabs submitted showed similar detection frequencies for EHV-1, *S. equi*, EHV-4, and ERBV between the first and second swab submitted (Table 3). Horses testing qPCR positive for ERBV or EIV had no third swabs submitted. However, all second swabs tested qPCR negative for EIV. Individual results for all horses testing qPCR positive for common respiratory pathogens are listed in Table 2.

## 3. Discussion

The study was designed around a unique opportunity to study the prevalence of common and less characterized respiratory pathogens in clinically healthy show horses being screened for EHV-1 following a multi-county outbreak of EHM in the spring of 2022. Regulatory testing provided samples from this population of healthy horses that would not be typically sampled from.

Clinically healthy sport horses were found to be qPCR positive for common respiratory pathogens, with a weekly frequency ranging between 5.0 and 17.9% and an overall frequency of 9.9%, suggesting silently shedding horses are present in sport horse populations. EHV-4, EIV, and ERBV had frequencies of qPCR-positive swabs within the previously reported ranges of 0–3.2%, 0–1%, and 0–7.1%, respectively, for different populations of healthy horses [3,5,6,7,8,9,10]. EHV-1 and *S. equi* had frequencies at or above the highest previously reported frequencies of 3.8% and 0.6%, respectively [2,12]. Increased frequency of EHV-1 qPCR-positive swabs was expected, as greater frequency of transmission occurred following the outbreak. *S. equi* was also found at a higher frequency than previously reported, though many cases occurred in clusters with 50% of *S. equi* qPCR positive horses originating from only two barns.

Less characterized respiratory pathogens had qPCR positive swabs at frequencies also within previously reported ranges of 3–77.4% for EHV-2; 3–84.4% for EHV-5; and 19.7–79% for *S. zooepidemicus* [3,5,6,7,8,9,10]. The high detection rates determined for EHV-2, EHV-5, and *S. zooepidemicus* reflect the ubiquitous occurrence of these respiratory pathogens in show horses. While less characterized respiratory pathogens were frequently detected in the study horses, their role in clinical disease has remained poorly understood. Positivity rates were relatively constant throughout the study weeks, indicating that these pathogens were unaffected by the outbreak conditions.

The peak in common respiratory pathogen frequency during week 3 was especially apparent in samples qPCR positive for EHV-1 and *S. equi*. Clusters of common respiratory pathogens in several barns suggested silent spread among horses living at the same barn. Furthermore, horses coming back from a show experiencing an outbreak may become the source of infection for resident horses. Such circumstances have recently been experienced during EHM outbreaks when show horses returning from events were responsible for secondary infections [14,15]. The decline in EHV-1 qPCR positive samples by week 5 suggests that silent transmission of viral respiratory pathogens reduced to expected pre-outbreak levels [2,12].

For EHV-1, EHV-4, *S. equi*, and EIV, clusters of at least two horses testing qPCR positive in the same week at the same barn occurred in at least two barns. The most notable of these was barn K with a cluster of five horses, all qPCR positive for *S. equi* during week 3. This is likely an example of horses spreading common respiratory pathogens throughout respective home barns, as four of the five horses were qPCR negative during week 2. This cluster of five horses also explains some of the inflated frequency of *S. equi* qPCR positive swabs compared to previously reported data. Two clusters of EIV occurred in barns G and H. Cases outside of these barns were not observed, indicating that EIV was likely localized to those barns. The subclinical shedding of EIV in the four clinically healthy show horses highlights the positive impact of vaccination against EIV on clinical disease. The show history of the positive horses and their specific location in the different barns was unknown to the investigators, precluding any conclusion regarding risk of infection from horses returning from the show. However, the clustering of cases following the return of show horses to the barn highlights the need to follow up with proper biosecurity protocols for traveling horses and the need to separate show from resident horses.

Without qPCR testing, silent shedders of respiratory pathogens remain unidentifiable. It is speculated that silently shedding horses contributed to the spread of EHV-1 during the EHM outbreak in Valencia, Spain, in 2021 and Ogden, USA, in 2011 [14,15]. While it was not the aim of this study to determine the effectiveness of the USEF return-to-competition protocols [16], the strategy of testing horses for EHV-1 had two apparent benefits; it identified silent EHV-1 shedders and prevented EHV-1 qPCR positive horses from competing at the next event. Biosecurity protocols [17], such as limiting horse to horse contact and temperature monitoring, instituted for the purpose of reducing transmission risk for horses returning from the shows, were not assessed in this study but should be considered for future investigation.

USEF and show requirements required at least one PCR test 28 days before the next show or two PCR tests 7 days apart. Repeated testing showed an overall decline in EHV-1 detection, which follows the transmission dynamic of most respiratory viruses. However, the frequency of detection between the first and second swab remained similar for *S. equi*, EHV-4 and ERBV, suggesting that these common respiratory pathogens were circulating at a low and constant frequency without causing clinical disease and outbreaks. This observation points toward the expected low rate of silent transmission of *S. equi*, EHV-4, and ERBV among adult healthy show horses in the spring.

Horses that were tested earlier in the sampling window (weeks 1–3) were often qPCR negative for EHV-1 with a follow-up test, with 8 of 10 horses with at least one follow-up test being qPCR negative for EHV-1 after being qPCR positive for EHV-1 during weeks 1–3. This suggests that testing clinically healthy horses for show regulations further from exposure and closer to the next event may provide a more accurate picture of recent infection. Additionally, the increase in EHV-1 qPCR positive horses during week 3 highlights the need for the testing of at-risk horses after an outbreak, as these horses would not have been identified without molecular testing and could have resulted in additional transmissions. The instituted quarantine period was successful at controlling the multi-county EHM outbreaks and bringing down the EHV-1 detection rate to a level apparently low enough to prevent further outbreaks.

Based on the nature of the study design, the authors identified various limitations. Demographic information, travel and vaccination history, and information pertaining to the management of the show horses returning home were not available to the authors. This information would have been important in order to determine various risk factors. Further, additional diagnostic laboratories were also receiving samples for the testing of EHV-1, and such samples were not included in this present study. Lastly, while testing extended over a period of 5 weeks, not all horses were tested at the same time intervals, and not all horses had multiple samples tested.

## 4. Materials and Methods

### 4.1. Study Population and Sample Collection

Nasal swabs were submitted for molecular EHV-1 testing through the Real-time PCR Research and Diagnostics Core Facility at the School of Veterinary Medicine, University of California at Davis, following multi-county EHM outbreaks in the spring of 2022 [18]. Samples came from clinically healthy sport horses, being kept at their respective home barn following USEF and show recommendations for return-to-competition protocols. USEF return-to-competition protocols were issued on Monday of week 1 (28 February 2022), and sampling data were collected over 33 days (28 February 2022 to 1 April 2022). Results were organized along the outbreak response timeline, with week 1 from 28 February 2022 to 4 March 2022, week 2 from 7 March 2022 to 11 March 2022, week 3 from 14 March 2022 to 18 March 2022, week 4 from 21 March 2022 to 25 March 2022, and week 5 from 28 March 2022 to 1 April 2022. As samples were submitted for diagnostic purposes only, signalment (age, breed, sex) was not included in the submission forms.

Rayon-tipped 6” swabs were collected from the rostral nasal passages by local veterinarians and shipped overnight to the laboratory for sample processing and EHV-1 analysis.

### 4.2. Sample Analysis

Nucleic acid extraction from nasal swabs was performed 24 h post-collection using an automated nucleic acid extraction system (QIAcube HT, Qiagen, Valencia, CA, USA) according to the manufacturer’s recommendations. The Quantitect Reverse transcription kit (Qiagen) was used for cDNA synthesis following the manufacturer’s directions with the following modifications. A volume of 10 µL of RNA was digested with 1 µL of gDNA WipeOut Buffer by incubation at 42 °C for three minutes and then briefly centrifuged. Then, 0.5 µL of Quantitect Reverse Transcriptase, 2 µL Quantitect RT buffer, 0.5 µL RT Primer Mix, 0.5 µL 20 pmol Random Primers (Invitrogen, Carlsbad, CA, USA) were added and brought up to a final volume of 20 µL and incubated at 42 °C for 40 min. The samples were inactivated at 95 °C for 3 min, chilled, and 60 μL of nuclease-free water was added to dilute the cDNA to an optimal concentration.

The initial swabs were tested for EHV-1 by qPCR, as previously reported [19], and the results were reported within 24 h of sample analysis to the submitting veterinary clinic. As part of the study project, purified nucleic acids were subsequently also tested for the presence of EIV, EHV-2, EHV-4, EHV-5, ERAV, ERBV, *S. equi,* and *S. zooepidemicus* using previously reported real-time TaqMan PCR assays [20,21,22]. To determine sample quality and efficiency of nucleic acid extraction, all the samples were analyzed for the presence of the housekeeping gene equine glyceraldehyde-3-phosphate dehydrogenase (eGAPDH), as previously described [19].

The frequency of respiratory pathogen detection from the study horses was evaluated using descriptive analyses.

## 5. Conclusions

This study showed that common and less characterized respiratory pathogens are present in a healthy sport horse population following multiple EHM outbreaks in the spring of 2022 in California. With the exception of EHV-1, all other respiratory pathogens were detected at a low frequency, independent of the testing time. This may reflect silent transmission in the population of clinically healthy sport horses. This was in sharp contrast to the detection of EHV-1, which, following the outbreaks, showed a steady decline. Overall, the results showed that the strategy of mandatory travel restrictions and EHV-1 testing was associated with eliminating additional EHM outbreaks and facilitating safe return to competition with no reported respiratory disease outbreaks following the subsequent shows in California.

## Figures and Tables

**Table 1 pathogens-11-01161-t001:** Summary of all positive swabs for common and less characterized respiratory pathogens categorized by week after the issuance of USEF guidelines. Total swabs from each week is indicated in the header.

	Week 1 (160 Swabs)	Week 2 (182 Swabs)	Week 3 (156 Swabs)	Week 4 (73 Swabs)	Week 5 (68 Swabs)	All Weeks (639 Swabs)
**Common resp. pathogens** EHV-1	5 (3.1%)	3 (1.6%)	10 (6.4%)	4 (5.5%)	2 (2.9%)	24 (3.8%)
*S. equi*	2 (1.3%)	3 (1.6%)	9 (5.8%)	3 (4.1%)	1 (1.5%)	18 (2.8%)
EHV-4	0	1 (0.5%)	5 (3.2%)	0	3 (4.4%)	9 (1.4%)
ERBV	1 (0.6%)	2 (1.1%)	4 (2.6%)	1 (1.4%)	0	8 (1.3%)
EIV	0	4 (2.2%)	0	0	0	4 (0.6%)
ERAV	0	0	0	0	0	0
**Total**	8 (5.0%)	13 (7.1%)	28 (17.9%)	8 (11.0%)	6 (8.8%)	63 (9.8%)
**Less characterized resp. pathogens**						
EHV-2	54 (33.8%)	60 (33.0%)	54 (34.6%)	20 (27.4%)	21 (30.9%)	209 (32.7%)
EHV-5	51 (31.9%)	69 (37.9%)	46 (29.5%)	23 (31.5%)	26 (38.2%)	215 (33.6%)
*S. zooepidemicus*	51 (31.9%)	49 (26.9%)	44 (28.2%)	18 (24.7%)	26 (38.2%)	188 (29.4%)
**Total**	111 (69.4%)	114 (62.6%)	93 (59.6%)	42 (60.3%)	43 (63.2%)	402 (62.9%)

**Table 2 pathogens-11-01161-t002:** Summary of 47 horses testing qPCR positive for at least one common respiratory pathogen categorized by week after the issuance of USEF guidelines. Horses were given a continuous number. Barn of origin was characterized by an individual letter.

Horse	Barn	Week 1	Week 2	Week 3	Week 4	Week 5
1	A	EHV-1				
2	B	*S. equi*				
3	C	*S. equi*	negative			
4	D	ERBV	ERBV			
5	E	EHV-1	negative			
6	F	EHV-1				
7	F	EHV-1				
8	F	EHV-1				
9	G		EIV			
10	G		EIV			
11	H		EIV	negative		
12	H		EIV, EHV-1	EHV-1		
13	H		negative	EHV-1		
14	I		*S. equi*			
15	J		*S. equi*	EHV-4, EHV-1	negative	
16	J		negative	EHV-4		
17	K		EHV-4	*S. equi*		
18	K		*S. equi*	*S. equi*, EHV-1	*S. equi*	
19	K		negative	*S. equi*		
20	K		negative	*S. equi*		
21	K		negative	*S. equi*		
22	L		EHV-1			negative
23	M		EHV1	EHV-4, EHV-1		negative
24	M		negative	EHV-4, EHV-1		negative
25	N		ERBV	ERBV		
26	O		negative	*S. equi*		
27	P			EHV-4		
28	P			ERBV		
29	Q			*S. equi*	negative	
30	Q			*S. equi*, EHV-1	negative	
31	Q			*S. equi*	negative	
32	R			EHV-1	EHV1	
33	S			ERBV		
34	T			ERBV		
35	U			EHV-1		negative
36	V			EHV-1		
37	W		negative	negative	*S. equi*	
38	X				ERBV	
39	Y				*S. equi*	
40	Z				EHV-1	negative
41	AA				EHV-1	negative
42	AA				EHV-1	negative
43	AB					EHV-1
44	AC					EHV-1, EHV-4
45	AC					EHV-4
46	AC					EHV-4
47	AD					*S. equi*

**Table 3 pathogens-11-01161-t003:** Summary of qPCR-positive swabs for each common respiratory pathogen determined based on number of swabs submitted. Swabs were submitted approximately one week apart. Total number of swabs per submission is indicated in the header.

	1st Swab (47 Swabs)	2nd Swab (27 Swabs)	3rd Swab (5 Swabs)
EHV-1	17 (36.2%)	7 (25.9%)	0
*S. equi*	10 (21.3%)	6 (22.2%)	2 (40.0%)
EHV-4	5 (10.6%)	4 (14.8%)	0
ERBV	6 (12.8%)	2 (7.4%)	Not available
EIV	4 (8.5%)	0	Not available
ERAV	0	0	0

## Data Availability

Data are available on request due to privacy restrictions.

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
