# Peer review of "Frequency of Detection of Respiratory Pathogens in Clinically Healthy Show Horses Following a Multi-County Outbreak of Equine Herpesvirus-1 Myeloencephalopathy in California"

_pathogens, 2022, doi:10.3390/pathogens11101161_

Round 1

Reviewer 1 Report

This work is important from the epidemiological point of view and helps to understand the dynamics of the spread of common and less frequent respiratory infections. Especially that the number of tested samples, including double samples from same animals, was quite large. Presented data also shows that certain travel restrictions are effective in eliminating these respiratory infections.

In my opinion, it would be interesting to report the number of animals both EHV-1 and EHV-2 positive, since according to earlier suggestions EHV-2 may play a role in EHV-1 reactivation from latency. 

Redesigning Table 2 would make it more clear.

This paper should be  short communication.

Author Response

This work is important from the epidemiological point of view and helps to understand the dynamics of the spread of common and less frequent respiratory infections. Especially that the number of tested samples, including double samples from same animals, was quite large. Presented data also shows that certain travel restrictions are effective in eliminating these respiratory infections.

In my opinion, it would be interesting to report the number of animals both EHV-1 and EHV-2 positive, since according to earlier suggestions EHV-2 may play a role in EHV-1 reactivation from latency. 

The reviewer brings up an interesting observation, which is the coinfection between EHV-2 and other respiratory pathogens. The investigators looked at the EHV-1 positive results and determined that 33.3% of EHV-1 qPCR-positive horses were also positive for EHV-2, while 66.6% of EHV-1 qPCR-positive horses tested qPCR-negative for EHV-2. The detection rate for EHV-2 within the study population ranged from 27.4 to 33.8% (average 32.7%), which is similar to the percentage of EHV-1/EHV-2 dually infected horses. Based on these results, it does not appear that coinfection with EHV-2 prevailed in horses that tested qPCR-positive for EHV-1.

Redesigning Table 2 would make it more clear.

The focus of Table 2 is to visually show the association between individual qPCR-positive results and origin of the horse and point towards clusters of infection. While the Table is long, it gives an easy to read overview of the positive horses and their origin. As an alternative, the authors could describe the findings in the results, which probably would make the result long and hard to understand.  

This paper should be short communication.

Because of the relevance of the results and the discussion, the authors would like to keep this article as a full lengths research article without having to shorten sections in order to comply with the format of a short communication.

Reviewer 2 Report

This is an interesting study however the manuscript is rather sloppy in areas and needs to be improved in several areas;-

 use of acronyms in title and not spelling out whn first used

poor detail of initial explanation of such terms a silent shedders, EHM outbreak 

Consider greater discussion of findings such as silent shedding EIV-this is not part of a normal

see attached file for greater detail 

Author Response

Frequency of detection of respiratory pathogens in clinically healthy  show horses following a multicounty EHM *outbreak in California *

The title has been changed to “Frequency of detection of respiratory pathogens in clinically healthy show horses following a multi-county outbreak of equine herpesvirus-1 myeloencephalopathy in California” as suggested by the reviewer.

Avoid acronyms in title and explain EHM in text prior to usage and provide more details in introduction or discussion of the “multi county EHM outbreak” -there is too much lack of detail for the readers in this phrase used in the text  Preface the term healthy horses with clinically healthy throughout.

Acronyms were spelled out the first time they were used in the manuscript. Further, as mentioned by the reviewer additional details were added regarding the multi-county outbreak and the term of “clinically healthy horses” has replaced the term “healthy horses” throughout the manuscript.

Abstract: Silently shedding horses have been indicated as a possible source of respiratory pathogen  outbreak, transmission, and spread.  Using nasal swabs from clinically healthy sport horses submitted for qPCR testing after an outbreak of equine herpesvirus-1 (EHV-1)Myeloencephalitis?  in the spring of 2022, this study aimed to identify the rate of silently shedding horses for common and less-characterized respiratory pathogens within the sport horse population to better understand their role in outbreaks. Swabs were collected during a required quarantine and testing period according to the United States Equestrian Federation (USEF) and show return-to-competition requirements

The abstract was edited as suggested by the reviewer.

Line 12 Please avoid starting the section with the term silently shedding rephrase to explain or introduce the term for an international readership before using it 

The sentence in Line 12 has been changed as follows: Actively shedding healthy horses have been indicated as a possible source of respiratory pathogen outbreak, transmission, and spread.

Line 37 p1 Recommend greater explanation for a scientific article than rather conversational language.  Subclinical infections with respiratory pathogens have been indicated as a possible explanation for outbreaks of disease [3] and horses silent shedding virus without signs of clinical disease (silent shedders)  have been identified 

The sentence has been changed as suggested by the reviewer.

Line 50 -1 p2  Please provide more detail than vague following multi county EHV-1 outbreaks during spring of 2022 in California

Additional information pertaining to the outbreak has been added.

Line 132 Whilst many pathogens described have recognised shedding from subclinical infections Perhaps some explanation on discussion of the detection of EIV in such horses may be beneficial to readers (ie sterile immunity vs clinical ) 

The comment from the reviewer is very pertinent and points towards silent infection of EIV in a population of vaccinated horses. A sentence was added in this regard and placed in line 143/144.

Line 223 Conclusion This study showed that common and less-characterized respiratory pathogens are  present in a healthy sport horse population following multiple EHM outbreaks in the spring of 2022 in California. With the exception of EHV-1, all other respiratory pathogens were detected  at a low frequency independent of the testing time This may reflec silent transmission in the population of healthy sport horses. This was in sharp contrast to the detection  of EHV-1, which following the outbreaks, showed a steady decline. Overall, the results showed that the strategy of mandatory travel restrictions and EHV-1 testingwas associated with eliminating additional EHM outbreaks and facilitating safe return-to-competition with no reported respiratory disease outbreaks following the subsequent shows in California.

The suggested changes were incorporated in the conclusion.